# The Effects of Stevia Consumption on Gut Bacteria: Friend or Foe?

**DOI:** 10.3390/microorganisms10040744

**Published:** 2022-03-30

**Authors:** Arezina N. Kasti, Maroulla D. Nikolaki, Kalliopi D. Synodinou, Konstantinos N. Katsas, Konstantinos Petsis, Sophia Lambrinou, Ioannis A. Pyrousis, Konstantinos Triantafyllou

**Affiliations:** 1Department of Nutrition and Dietetics, Attikon University General Hospital, 12462 Athens, Greece; kastiare@med.uoa.gr (A.N.K.); maroullanikolaki@gmail.com (M.D.N.); kall.synodinou@gmail.com (K.D.S.); kostasktss1@gmail.com (K.N.K.); kostas.petsis@hotmail.com (K.P.); sophialambrinou@gmail.com (S.L.); ipyrousis@gmail.com (I.A.P.); 2Institute of Preventive Medicine Environmental and Occupational Health Prolepsis, 15125 Athens, Greece; 3Medical School, University of Patras, 26504 Patras, Greece; 4Hepatogastroenterology Unit, 2nd Department of Propaedeutic Internal Medicine, Attikon University General Hospital, Medical School, National and Kapodistrian University of Athens, 12462 Athens, Greece

**Keywords:** *Stevia rebaudiana*, stevioside, gut microbiota, bacteria, fecal flora

## Abstract

Stevia, a zero-calorie sugar substitute, is recognized as safe by the Food and Drug Administration (FDA) and the European Food Safety Authority (EFSA). In vitro and in vivo studies showed that stevia has antiglycemic action and antioxidant effects in adipose tissue and the vascular wall, reduces blood pressure levels and hepatic steatosis, stabilizes the atherosclerotic plaque, and ameliorates liver and kidney damage. The metabolism of steviol glycosides is dependent upon gut microbiota, which breaks down glycosides into steviol that can be absorbed by the host. In this review, we elucidated the effects of stevia’s consumption on the host’s gut microbiota. Due to the lack of randomized clinical trials in humans, we included in vitro using certain microbial strains and in vivo in laboratory animal studies. Results indicated that stevia consumption has a potential benefit on the microbiome’s alpha diversity. Alterations in the colonic microenvironment may depend on the amount and frequency of stevia intake, as well as on the simultaneous consumption of other dietary components. The anti-inflammatory properties of stevioside were confirmed in vitro by decreasing TNF-α, IL-1β, IL-6 synthesis and inhibiting of NF-κB transcription factor, and in vivo by inhibiting NF-κB and MAPK in laboratory animals.

## 1. Introduction

*Stevia rebaudiana* (Bertoni) is a natural, non-caloric sweetener (~200–400 times higher than sucrose). Its sweet taste occurs from the steviol glycosides, especially stevioside and rebaudioside A (REB-A) together with rebaudioside C and dulcoside A [1,2]. Until now, more than 40 steviol glycosides have been identified, which are classified as ent-kaurene-type diterpenes with sugar fractions attached to the aglycone steviol. Steviol glycosides cannot be broken through enzymes such as pancreatic α-amylase, pepsin, and pancreatin found in saliva and gastric secretions, and pass intact through the upper gastrointestinal tract where finally they are hydrolyzed by intestinal bacteria to steviol [3,4,5]. *Bacteroides* hydrolyze stevioside and REB-A to steviol, while other bacteria such as *Lactobacilli*, *Bifidobacteria*, *Clostridia*, *Coliforms*, and *Enterococci* cannot [6]. Absorbed steviol via the portal vein reaches the liver, is metabolized to steviol glucuronide, and is excreted in the urine [3,4]. According to the Commission Regulation (EU) 1131/2011, the acceptable daily intake (ADI) for steviol equivalents is 4 milligrams (mg) per kilogram of body weight [7].

Stevia’s superiority against sucrose and artificial sweeteners was confirmed many years ago. Given its safety, studies revealed beneficial properties for human health [8]. In vitro and in vivo, stevia showed anti-viral effects [8,9], immunomodulatory activity, and anti-inflammatory properties by inhibiting the activation of nuclear factor-kappa B (NF-κB), mitogen-activated protein kinase (MAPK) signaling, and the release of proinflammatory cytokines [10,11,12,13]. In rats, stevioside showed antiglycemic effects by increasing insulin secretion, decreasing plasma glucose concentrations, and suppressing glucagon levels, although the underlying mechanism has not been clarified yet [14,15]. Beyond improved insulin signaling and the antioxidant effect in the adipose tissue and the vascular wall, stevia also significantly reduced blood pressure levels (systolic and diastolic), whereas it stabilizes the atherosclerotic plaque and inhibits its further development [15,16]. Other studies on rodents found that stevia-derived compounds reduce hepatic steatosis [17] and ameliorate liver and kidney damage [8,18]. In vitro, steviol glycoside derivatives were found to possess antiproliferative and anticancer activity, via the mitochondrial apoptotic pathway, in several cancer cell lines, including breast [19,20], prostate [21], gastric [22], and colon cancer cell lines [23].

Nowadays, the gut microbiota is considered an organ that regulates metabolism, cellular immune response, and contributes to the host’s health. The human gut microbiota shows a wide variation, but its within-individual variation is relatively stable over time, with Firmicutes and Bacteroidetes representing 90% of the dominant phyla [24].

An imbalance in intestinal bacteria leads to dysbiosis, and animal and human studies have demonstrated that diet can rapidly influence its composition and function [25]. So far, stevia has become extremely popular because it is derived from plants and is healthier than other artificial sweeteners. The increased use of stevia as a safe sweetener for the host raises questions about whether its consumption is safe for intestinal bacteria. Several studies, mainly in laboratory animals, have identified potential side effects of stevia over the last few decades. Stevia metabolism is dependent upon gut microbiota and microbial enzymes can break down these glycosides into steviol that can be absorbed by the host. However, the effects of stevia on the gut microbiota need to be further studied [26]. In this review, we aim to investigate the effects of stevia’s intake on the host’s gut microbiota.

## 2. Materials and Methods

We performed a literature search in PubMed for articles in the English language. We used evidence from original articles, excluding reviews, abstracts, conference presentations, editorials, and study protocols. The search was based on the terms “stevia, gut”, “stevia, microbiota”, “stevia, fecal flora”, and “gut, rebaudioside A”. Studies identified by a manual search of the reference lists were included (Figure 1).

We assessed preclinical studies (in vivo, ex vivo) examining the use of stevia during the last decade. Randomized clinical trials have not been performed yet in humans, and to date, there is no evidence about stevia’s meaningful impact on the composition or function of the gut microbiota [27,28,29,30].

## 3. The Effects of Stevia and Steviol Glycosides on Bacterial Growth

### 3.1. In Vitro Studies

The summary of the evidence of in vitro studies is shown in Table 1. Li et al. studied the effect of stevia on bacterial communities. Experiments were performed with two Gram-negative pathogens (*Escherichia coli* O157:H7, *Salmonella typhimurium* ATCC 13311), two Gram-positive pathogens (*Staphylococcus aureus* CGMCC 26001, *Listeria monocytogenes* CMCC 54007), and two probiotics (*Bifidobacterium longum* ATCC 15707, *Lactobacillus plantarum* ACCC 11095). The results showed no effect on the growth of the pathogens *E. coli*, *S. typhimurium*, *L. monocytogenes*, nor in the probiotic species *B. longum* and *L. plantarum,* whereas significant reduction was observed for *S. aureus CG* (*p* < 0.01) in a concentration-dependent manner [27]. Similarly, QP Wang et al. indicated that REB-A exerts a selective bacteriostatic effect on gut flora, significantly inhibiting the growth of *E. coli* HB101, but not *E. coli* K-12 [28].

Based on the theory that prophage induction in bacteria may result in the horizontal transfer of genes to other bacterial strains or species, researchers tested three gut bacteria, *B. thetaiotaomicron*, *S. aureus*, and *Enterococcus faecalis*, for their response to stevia as a prophage inductor. They found that stevia increased virus-like particles (VLPs) detected at 410% and 321% from *B. thetaiotaomicron* and *S. aureus*, respectively [31]. The abundance of terpenes (naturally occurring chemical compounds found mainly in plants) is possibly responsible for the antimicrobial properties of stevia [31], with a potential mechanism of action to be related to the rupture or dysfunction of their cell membrane. Given that previous works showed strain-specific bacteriostatic effects of stevia, it is interesting to note that studies agreed with its effectiveness against *S. aureus* but not against *E. faecalis* [32,33].

Mahalak et al. performed an experiment comparing changes to the gut microbiota in the feces of a healthy donor when exposed to steviol glycosides and erythritol. Results showed that common gut bacteria have a limited growth response to stevia components. The presence of steviol had a statistically significant increase in growth compared with the control only for *Bacteroidetes thetaiotaomicron.* The typical stabilized human gut microbiota remained the same, with the *Bacteroidaceae* family being dominant, followed by *Lachnospiraceae*, *Fusobacteraceae*, and *Eubacteraceae* [34]. Gerasimidis et al. measured the effect of stevia using human microbiome batch fermentations and observed no significant differences in the growth of *Bacteroides/Prevotella, Bifidobacterium, Blautia coccoides, Clostiridium leptum,* and *E. coli* [35]. These results were consistent with the work of Kunová et al., who highlighted the lack of prebiotic effect of REB-A and steviol glycosides. Eight *Bifidobacteria* and seven *Lactobacilli* were cultured and tested for their ability to grow in the presence of REB-A and steviol glycosides. The growth of some *Bifidobacteria species (Bifidobacterium bifidum CCDM 559, Bifidobacterium breve CCDM 562,* and *Bifidobacterium adolescentis AVNB3- P1)* was higher than others, but no significant changes were detected. Among *Lactobacilli, Lactobacillus mucosae SP1TA2-P1* grew the most. Overall, neither *Bifidobacteria* nor *Lactobacilli* can substantially use steviol glycosides as a substrate, indicating their very poor fermentation [30].

On the contrary, Denina et al. claimed that stevia glycosides—stevioside and REB-A—inhibit *Lactobacillus reuteri* growth in a strain-dependent manner [29]. In another prototype trial, researchers evaluated the effects of stevia on the bacterial ability to detect and respond to cell population density by gene regulation (quorum sensing, QS). QS is an essential communication system (intra- and inter-bacterial) that enables many features of bacterial community behavior to be regulated. Experiments were conducted with a recombinant bioluminescent *E. coli* K802NR-pSB1075 and the lasRI gene from *Pseudomonas aeruginosa.* Results showed that stevia might lead to microbial imbalance, disrupting the communication between Gram-negative bacteria in the gut via either the LasR or RhlR receptor proteins of *P. aeruginosa*. However, even if stevia inhibits these pathways, it cannot kill off the bacteria [36]. Table 1.

### 3.2. In Vivo Studies

Researchers hypothesized that stevia could correct high-fat-diet-induced glucose intolerance by altering the gut microbiota, but results in a murine model highlighted no impact on glucose intolerance nor protection from high-fat-diet-induced changes. The significant increase in *Firmicutes*/*Bacteroidetes* ratio correlated with the high-fat diet and obesity [26]. In contrast to this publication, Yu et al. investigated the effects of different supplementation levels of stevia residues in high-fiber diets on the fecal bacteria of pregnant mammalians. It is known that high-fiber diets can promote the abundance of beneficial bacteria *Bifidobacteria* and *Lactobacilli* and improve intestinal balance. The parallel stevia-residue supplementation significantly increased the beneficial and reduced the harmful bacteria, while the optimal supplementation level of the stevia residue was 30% [37]. Another trial evaluated the dose-dependent effects of REB-A (low (0.5 mg/mL) and high dose (5.0 mg/mL)), and indicated that the different doses did not affect the growth of *Enterobacteria* and *Lactobacilli* nor alter the microbial diversity but might have changed the number of some bacterial genera [27].

Reimer et al. attempted to prove that prebiotic consumption can reverse the potential adverse effects of stevia. REB-A reduces the relative abundance of *Bifidobacteriaceae*—the “health-promoting” bacteria—but increases *B. thetaiotaomicron*, which stimulates Paneth cells and promotes intestinal angiogenesis. A significantly greater abundance of these taxa was induced in rats on prebiotics compared to that in the non-probiotic group. Stevia and prebiotic consumption protected from alterations in gut microbiota composition observed in the group with REB-A consumption only [38]. The increasing evidence that gut microbiota in offspring is shaped in part from maternal diet led the scientific community to investigate the role of stevia during the prenatal period, pregnancy, and lactation. Thus, they observed alterations of fecal microbiota in dams and offspring fed with stevia correlated with a greater risk for metabolic syndrome (increased *Porphyromonadaceae*), and type-2 diabetes (increased *Sporobacter*) [39]. In continuation of studying the possible mechanisms by which maternal consumption of stevia increases the risk of altered gut microbiota in offspring, investigators recently reconstructed the most significant alterations of the cecal microbiome in the offspring of obese dams consuming a high fat/sucrose (HFS) diet with or without stevia. Stevia had limited influence on the overall structure of cecal microbiota in dams but induced significant alterations in offspring. Consequently, maternal consumption contributes to the metabolic changes in the offspring who were never directly exposed to stevia [40].

Given that the gut–brain axis plays a crucial role in the etiology of mental illness and cognitive and memory disorders, de la Garza et al. indicated that maternal gut dysbiosis deteriorates learning procedures and leads to memory loss susceptibility in adult male offspring rats. A maternal high-stevia diet induced the upregulation of *Bacteroidales* and *Clostridiales*, leading to memory loss and cognitive problems in offspring lasting up to adulthood, while the changes found in these phyla were independent of their body weight gain [41]. A summary of the above discussed studies is shown in Table 2.

## 4. The Effects of Stevia and Steviol Glycosides on Microbial Diversity

Species diversity is a measure of “health” in an ecosystem. Total species diversity in a landscape, with regards to spatial scale, is determined by two different indicators: the average species diversity at the local level (alpha diversity) and the differentiation among local sites (beta diversity). More specifically, alpha diversity is defined as “the average species diversity in a particular area or habitat”, and beta diversity as “the diversity of species between two habitats or the measure of similarity or dissimilarity of two regions” [42]. In our review, we detected eight studies measuring alpha diversity, using a variety of different indices (Shannon index, Simpson index, Pielou’s evenness, Operational Taxonomic Units, Chao1 richness, and Faith’s Phylogenetic Diversity Index) [27,29,34,35,37,38,39,41]. Furthermore, we identified three studies evaluating beta diversity [34,38,39] (Table 3). Stevia consumption did not change beta diversity significantly in all studies [34,38,39]. The results regarding the effect of stevia in alpha diversity were contradictory. Alpha diversity did not significantly differ in three studies for stevia and control groups [28,37,38]. On the contrary, four studies—including the only study with a sample of human feces fermented in batch cultures [35]—showed significantly higher alpha diversity in the intervention group as compared to the controls [27,34,35,39]. De la Garza et al. assessed feces from male dams fed with a high-stevia diet [41] and reported a significantly higher alpha diversity index in controls than in the stevia group during breastfeeding, but the difference during adulthood was non-significant. The aforementioned studies indicate a potential benefit of stevia consumption in alpha diversity, but the lack of human trials does not allow extractions of safe conclusions.

## 5. Conclusions

Herein, we reviewed fourteen studies. Some of them have shown beneficial or no harmful effects of stevia and its components on gut microbiota, while others indicated harmful effects, potentially, using in vitro and in vivo models (Table 4). We must note that four studies using obesity-induced lab animals examined potential adverse effects of stevia supplementation on the beneficial microbial communities. The authors concluded that this effect was rather due to HFS diets than to stevia. Only four studies showed that stevia is harmful for gut microbiota [29,31,36,38], while one study showed that REB-A and stevioside might interrupt the Gram-negative bacterial communication [36]. In another study, both glycosides impaired the growth of six *Lactobacillus reuteri* strains in vitro [29].

Among the reviewed preclinical studies, we observed several confounding factors, for example, different dietary interventions, small sample size (e.g., one subject per group), no control group, or the use of different end products and doses. Furthermore, we should note that even the administered doses in the majority of studies were lower than ADI and they may not be relevant to humans (different gastrointestinal physiology and function). Although the distal gut microbiota of mice and humans harbor the same bacterial phyla, most bacterial genera and species found in mice are not present in humans. In vitro studies are significantly limited in biological relevance due to limitations in directly extrapolating tested concentrations to human exposure levels. Although stevia may change the colonic microenvironment, this effect seems to depend on the amount, the frequency of intake, and the other dietary components of the food, a fact that could be confusing [3]. Even if most of the studies (Table 3) show promising results regarding its potential benefits to modulate gut microbiota, study design limitations induce difficulties in comparing and interpreting the results. Besides the aforementioned effects, the anti-inflammatory properties of stevioside were confirmed in vitro in colonic epithelial cells (Caco-2), where both stevioside and steviol decreased TNF-α, IL-1β, and IL-6 synthesis, and inhibited NF-κB (p65) signaling pathway [43], and in vivo by inhibiting NF-κB and MAPK in colon tissues of Dextran Sulphate Sodium-induced colitis in mice [12] and intestinal mucosal damage of broiler chickens [13]. Although the data are more or less contradictory, we may speculate that stevia’s substances might mimic probiotic action protecting from inflammatory process and dysbiosis (Figure 2).

Stevioside and its metabolite steviol also have an inhibitory effect on inflammatory cytokine production via attenuating the IκBα/NF-κB signaling pathway (canonical pathway) and the MAPK signaling pathway. They decrease the IKKβ ability to phosphorylate the NF-κB inhibitor IκBα, which would result in the dissociation of the ΙκBα from ΝF-κB, the ubiquitination of the ΙκBα, and the proteasome degradation of ΙκBα. Stevioside and steviol also inhibit the MAPK signaling pathway by attenuating the phosphorylation of p38, ERK and JNK proteins and abrogate the activation of NF-κB transcriptional factor. Therefore, they inhibit the subsequent phosphorylation of NF-κB and its translocation to the nucleus [10,11]. There is evidence that several probiotic strains can modulate the Nf-κB pathway and MAPK pathway in the same sites [44,45].

We recognize that we cannot easily extrapolate the results of these studies in humans, while germ-free mice models receiving human fecal transplantation could be a model to examine a gut microbial profile representative of humans. Further research is required to provide evidence of the role of stevia on the human gut microbiota.

## Figures and Tables

**Figure 1 microorganisms-10-00744-f001:**
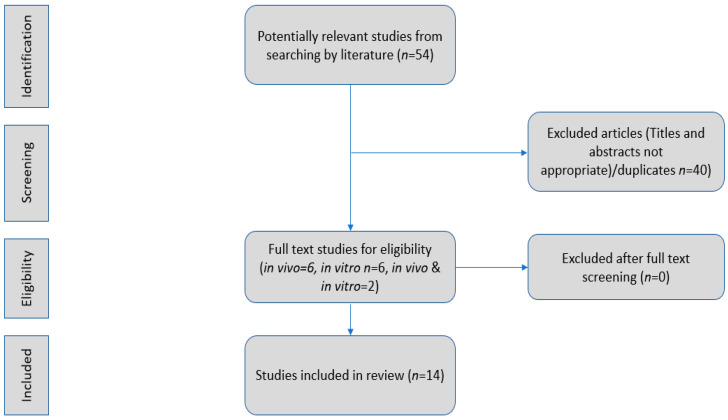
Flow chart. Identification and selection of the studies.

**Figure 2 microorganisms-10-00744-f002:**
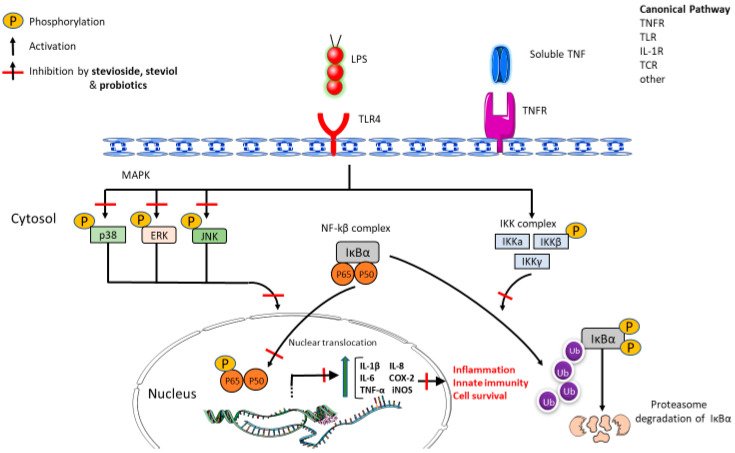
The anti-inflammatory effect of stevia glycosides through inhibition of transcription factor NF-κB and mitogen-activated protein kinase (MAPK). Figure created using Servier Medical Art (www.servier.com; accessed on 28 January 2022) under a Creative Commons Attribution 3.0 Unported license.

**Table 1 microorganisms-10-00744-t001:** In vitro studies presenting the effects of stevia and steviol glycosides on bacterial growth.

Reference	Strains	Intervention	Control	Beneficial or No Effect on Bacterial Populations Growth	Adverse Effects on Bacterial Populations Growth
Markus et al. 2020	*E. coli* K802NR-pSB1075	REB-A Stevioside Steviol	LB		Possible interruption of Gram-negative bacterial communication
Li et al. 2014	*E. coli* O157:H7 *S. typhimurium* ATCC 13311 *S. aureus* CGMCC 26001 *Listeria monocytogenes* CMCC 54007 *Bifidobacterium longum* ATCC 15707 *Lactobacillus plantarum* ACCC 11095	REB-A	Saline buffer	No effect *E. coli,* *S. typhimurium,* *Listeria monocytogenes* *Bifidobacterium longum,* *Lactobacillus plantarum* Reduced *S. aureus CG **	
Wang et al. 2018	*E. coli* HB101 and K-12	REB-A	LB with agar	Reduced *E. coli* HB101 No effect *E. coli* K-12	
Deniņa et al. 2014	*Lactobacillus reuteri* (six strains)	REB-A Stevioside	Acetic acid and lactic acid		Inhibit *Lactobacillus reuteri* strains
Boling et al. 2020	*B. thetaiotaomicron* VPI-5482 *E. faecalis* *S. aureus*	REB-A	None	Reduced *S. aureus*	Increased *E. faecalis* Reduced *B. thetaiotaomicron VPI-5482*
Gerasimidis et al. 2020	*Bacteroides/Prevotella* *Bifidobacterium* *Blautia coccoides* *C. leptum* *E. coli*	Stevia	None	No effect *Bifidobacterium,* *Blautia coccoides,* *Bacteroides/Prevotella,* *C. leptum,* *E. coli*	
Mahalak et al. 2020	Human gut microbiota	Steviol glycosides + erythritol	None	Increased *B. thetaiotaomicron* No effect *E. coli* *Enterococcus caccae* *L. rhamnosus* *Ruminococcus gauvreauii Bacteroides galacturonicus*	
Kunová et al. 2014	*Bifidobacteria* *(longum subsp. CCDM 219,* *animalis subsp. lactis CCDM 94,* *dentium CCDM 318,* *breve CCDM 562,* *bifidum CCDM 559,* *adolescentis AVNB3-P1,* *bifidum JKM,* *bifidum JOV)* *Lactobacilli* (*brevis CCDM 202,* *delbrueckii subsp., bulgaricus CCDM 66,* *acidophilus CCDM 151,* *paracasei subsp. CCDM 212,* *mucosae SP1TA2)*	Stevioside and REB-A (purity ≥ 95% of steviol glycosides), medium containing REB-A	MRS broth and D-glucose	Increased *Lactobacillus mucosae SP1TA2* Increased *Bifidobacterium bifidum CCDM 559,* *Bifidobacterium breve CCDM 562,* *Bifidobacterium adolescentis AVNB3- P1* No effect *B. longum subsp. longum CCDM 219* *B. animalis subsp. lactis CCDM 94* *B. dentium CCDM 318* *B. bifidum JKM* *B. bifidum JOV* No effect *L. brevis CCDM 202* *L. delbrueckii subsp. bulgaricus CCDM 66* *L. acidophilus CCDM 151* *L. paracasei subsp. paracasei CCDM 212*	

* *p*-value *p* < 0.05.

**Table 2 microorganisms-10-00744-t002:** In vivo studies presenting the effects of stevia and steviol glycosides on bacterial growth.

Reference	Type of Study	Model/Samples	Intervention	Control	Beneficial or No Effect on Bacterial Populations Growth	Adverse Effects on Bacterial Populations Growth
Becker et al. 2020	Preclinical RCT	Mice (feces)	HFS + stevia	Saccharin		Increased *Firmicutes/Bacteroidetes* ratio
Li et al. 2014	Preclinical RCT	Mice (feces *Enterococci* *Enterobacteria Lactobacilli)*	Low dose REB-A (0.5 mg/mL) High dose REB-A (5.0 mg/mL)	None	Increased *Lactobacilli* (high dose only) No effect *Enterococci* *Enterobacteria*	
Nettleton et al. 2019	Preclinical RCT	Rats (feces *Bifidobacteriaceae* *Enterobacteriaceae*)	REB-A and REB-A + prebiotic	Water	Increased *A. muciniphila* (in both groups)*,* *Bacteroides goldsteinii* (REB-A group) *B. thetaiotaomicron* (REB-A group) (correlated with intestinal angiogenesis) Reduced *Clostridiales* family XIII (in both groups), *Lactobacillus intestinalis* (REB-A group)	Reduced *Ruminococcaceae* UCG 005 (in both groups), *Bifidobacteriaceae* (REB-A group)
Nettleton et al. 2020	Preclinical	Obese rats during pregnancy and lactation and their offspring (feces)	HFS + REB-A	Lean rats during pregnancy and lactation and their offspring: control diet	Obese rats and offspring Reduced *Bifidobacterium*	Obese rats Increased *C. leptum* Obese rats and offspring Increased *Porphyromonadaceae* (metabolic syndrome development) *Sporobacter* (type-2 diabetes development) *Enterobacteriaceae* (proinflammatory)
Wang et al. 2022	Preclinical RCT	Obese rats during pregnancy and lactation and their offspring (Distal jejunum, ileum tissue, cecal digesta)	HFS + stevia	Rats during pregnancy and lactation: HFS + water Offspring: control diet		Increased *14_Bacteroidaceae* unclassified Reduced *A. muciniphila* *Limosilactobacillus reuteri*
de la Garza et al. 2022	Preclinical RCT	Rats during pregnancy and lactation and their male offspring (feces)	In prenatal period: cafeteria diet. In gestation and lactation: Stevia + control diet Offspring: control diet	Control diet	Maternal and male offspring group Reduced *Bacteroidetes, Cyanobacteria* Increased *Firmicutes* *Elusimicrobia* (correlated with decreased blood glucose levels)	Maternal and male offspring group Increased *Firmicutes*/*Bacteroidetes* ratio, *Bacteroidales* *Clostridiales* (contribute to cognitive dysfunction)
Mahalak et al. 2020	Preclinical	Monkey *(Cebus apella)* (feces)	Steviol glycosides + erythritol	-	No effect in the microbial community	
Yu et al. 2020	Preclinical RCT	Pregnant sows (feces)	Corn–soybean-meal diets + stevia residue 20%, 30%, 40%	Control diet	Increased *Lachnospiraceae_XPB1014, Christensenellaceae_R-7_ Ruminococcaceae_UCG-005* Reduced *Treponema_2*	

REB-A: Rebaudioside A; *E. coli*: *Escherichia coli*; LB: Lysogeny broth; MRS broth: de Man, Rogosa and Sharpe broth; HFS: high fat/high sucrose diet; E. faecalis: Enterococcus *faecalis*; *S. aureus*: *Staphylococcus aureus*; *P. aeruginosa*: *Pseudomonas aeruginosa*; *B. thetaiotaomicron*: *Bacteroides thetaiotaomicron*; *L. rhamnosus*: *Lactobacillus rhamnosus*; *B. bifidum*: *Bifidobacterium bifidum*; *C. leptum*: *Clostridium leptum*; *A. muciniphila*: *Akkermansia muciniphila*; *S. typhimurium*: *Salmonella typhimurium*.

**Table 3 microorganisms-10-00744-t003:** The effects of stevia and steviol glycosides on microbial diversity.

Reference	Target Group	Evaluate	Alpha Diversity	Beta Diversity
Li et al. 2014	Mice	a-diversity measures: Richness, H’^A^ and SE	(1) DGGE using V3 universal primers or using *Enterobacteriaceae* primers: NS differences in Richness, H’^A^, H’^A^_MAX_ & SE (2) DGGE using *Lactobacilli* primers: Significant higher Richness & H’^A^_MAX_ in high SG compared with CG (*p* < 0.05): - Richness: 11.2 ± 0.84 (SG) vs. 8.9 ± 0.84 (CG) - H’^A^_MAX_: 2.41 ± 0.08 (SG) vs. 2.28 ± 0.08 (CG)	-
Nettleton et al. 2019	Mice	(1) a-diversity measures: Chao, H’^A^ and Simpson (2) b-diversity measures: NMDS	NS difference in alpha diversity measures between CG and SG	NS difference in beta diversity measures between CG and SG
Nettleton et al. 2020	Mice	(1) a-diversity measures: H’^A^, and Simpson (2) b-diversity measures: weighted and unweighted UniFrac distances	Significantly higher a-diversity measures in SG compared to CG	NS difference in beta diversity measures between CG and SG
Wang et al. 2018	Mice	a-diversity measure: H’^A^	NS difference in alpha diversity measures between CG and sucralose in normal chow or HFD-fed mice	-
Gerasimidis et al. 2020	13 healthy volunteers	a-diversity measures: OTUs, Chao, Rarefied richness, H’^A^, J’	Addition of stevia significantly increased H’^A^, J’ and Rarefied richness (compared to CG)	-
de la Garza et al. 2022	Mice (male)	a-diversity measure: H’^A^	(1) Significantly higher H’^A^ in CG compared to SG during breastfeeding. (2) NS difference in H’^A^ during adulthood period (CG vs. SG). (3) NS difference in H’^A^ between breastfeeding and adulthood period in SG.	-
Mahalak et al. 2020	In vitro	(1) a-diversity measures: Species Richness, H’^A^, and Fa (2) b-diversity measures: weighted and unweighted UniFrac distances	NS difference in alpha diversity measures over time between CG, Erythritol group and SN Stevia group	No consistent pattern was observed between each group
	1 volunteer in vivo		Consumption of SN Stevia & Erythritol increased alpha diversity measures significantly over time (*p* < 0.05)	NS difference in beta diversity measures over time
Yu et al. 2020	Sows	a-diversity measures: OTUs, Sobs, Chao1, Ace, H’^A^, Simpson, Coverage index	NS difference in alpha diversity measure between CG and experimental groups fed with stevia residue	

NS: No significant difference (*p* > 0.05); HFD: high-fat diet; CG: control group; SG: Stevia group; H’^A^: Shannon index OR Shannon’s diversity index OR Shannon–Wiener index (same); SE: Shannon evenness index; J’: Pielou’s evenness; OTUs: operational taxonomic units; Chao: Chao1 richness; Fa: Faith’s Phylogenetic Diversity Index; NMDS: Nonmetric multidimensional scaling on a Bray–Curtis dissimilarity matrix.

**Table 4 microorganisms-10-00744-t004:** The effects of stevia glycosides on certain beneficial and harmful bacteria growth in in vitro and in vivo studies, without any dietary intervention.

Ref	Beneficial Effect	Harmful Effect
Beneficial Strains Growth	Suppression of Pathogens	Suppression of Beneficial Strains	Pathogen Growth
[5]	*Lactobacilli*	*S. aureus CG*		
[6]	*A. muciniphila Bacteroides goldsteinii* *B. thetaiotaomicron*	*Clostridiales* family XIII*Lactobacillus intestinalis*	*Ruminococcaceae* UCG 005*Bifidobacteriaceae*	
[7]		*E. coli* HB101		
[8]			*Lactobacillus reuteri* (six strains)	
[10]		*S. aureus*	*B. thetaiotaomicron VPI-5482*	*E. faecalis*
[14]	*B. thetaiotaomicron*			
[15]	*Lactobacillus mucosae SP1TA2**Bifidobacterium bifidum CCDM 559*,*Bifidobacterium breve CCDM 562*,*Bifidobacterium adolescentis AVNB3-P1*

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
