# Peer review of "The Effects of Stevia Consumption on Gut Bacteria: Friend or Foe?"

_microorganisms, 2022, doi:10.3390/microorganisms10040744_

Round 1
Reviewer 1 Report
The literature review is aimed to observe the effects of stevia consumption on gut bacteria, the reported results regarding in vivo and in vitro studies are discordant on the beneficial or harmful effect, for this reason more studies are needed to deeply understand this interaction.
The review is well written and easy to follow, the data are well presented.
Minor revisions:
-Line 37-38-113-114-142 etc, write taxonomic names in italics
-Line 59, choose today or nowadays
-Figure 1, if possible, maximize the image quality
-Line 191, evaluating beta diversity, the ratio between
Author Response
Thank you for your comments; we have corrected all four issues that you detected in the revised manusript
Reviewer 2 Report
The review paper by Kasti et al. about the effect of Stevia consumption on Gut microbiota is important. I have some comments before it could get accepted:
- The authors should elaborate more in the introduction about the Stevia rebaudiana plant and its metabolized products.
- Figure 1: only 14 articles were included. This was based on what? How many were in vitro and how many were in vivo?
- The vertical boxes in figure.1 (included, eligibility , screening, identification) need restructuring and they don’t match the chart.
- Line 113 and 114, bacterial family names should be italic.
- Typo line 157 page 6 (non-probiotic)
- In page 8, give a brief description about beta diversity. The definition ( -the ratio between alpha di-diversity and regional diversity) was not very clear. What is the difference between alpha and beta?
- Line 232 and 234 in vitro and in vivo (italic)
Author Response
Reviewer 2. The review paper by Kasti et al. about the effect of Stevia consumption on Gut microbiota is important. I have some comments before it could get accepted:
The authors should elaborate more in the introduction about the Stevia rebaudiana plant and its metabolized products.
Response: In the introduction, page 1, row 35 we added: together with rebaudioside C and dulcoside A. Till now, have been identified more than 40 steviol glycosides, which are classified as ent-kaurene-type diterpenes with sugar fractions attached to the aglycone steviol. Steviol glycosides cannot be broken through enzymes like pancreatic α-amylase, pepsin and pancreatin found in saliva and gastric secretions and pass intact through the upper gastrointestinal tract while finally are hydrolyzed by intestinal bacteria to steviol.
Figure 1: only 14 articles were included. This was based on what? How many were in vitro and how many were in vivo?
Response: You may see the literature search in the first paragraph of the Materials and methos section
The vertical boxes in figure.1 (included, eligibility, screening, identification) need restructuring and they don’t match the chart.
Response: corrected. Figure 1 was replaced with a higher quality flow chart, while we highlighted the specific number of in vitro and in vivo studies.
Line 113 and 114, bacterial family names should be italic.
Typo line 157 page 6 (non-probiotic)
Response: corrected
In page 8, give a brief description about beta diversity. The definition ( -the ratio between alpha di-diversity and regional diversity) was not very clear. What is the difference between alpha and beta?
Response: We added: Species diversity is a measure of “health” in an ecosystem. Total species diversity in a landscape, with regards to spatial scale, is determined by two different indicators, the average species diversity at the local level (alpha diversity) and the differentiation among local sites (beta diversity). More specifically, alpha diversity is defined as “the average species diversity in a particular area or habitat”, and beta diversity as “the diversity of species between two habitats or the measure of similarity or dissimilarity of two regions”.
Line 232 and 234 in vitro and in vivo (italic)
Response: corrected